# A Cross-Sectional Analysis of Health Literacy and Compliance to Treatment in Organ Transplant Recipients

**DOI:** 10.3390/jcm12030977

**Published:** 2023-01-27

**Authors:** Sun Hyoung Bae, Jung Jun Lee, Sun Young Son, Hee Young Kim, Man Ki Ju

**Affiliations:** 1Research Institute of Nursing Science, College of Nursing, Ajou University, Suwon 100204, Republic of Korea; 2Department of Surgery, Gangnam Severance Hospital, Yonsei University College of Medicine, Seoul 03722, Republic of Korea

**Keywords:** transplantation, health literacy, adherence

## Abstract

This study was conducted to determine the correlations between health literacy, transplant effects, and compliance to treatment in organ transplant recipients and to identify the factors influencing compliance to treatment. The participants (*n* = 130; males = 66.9%; mean age = 56.4 years) were organ transplant recipients visiting an organ transplantation center in Seoul, South Korea. The regression model explained 32% of the variance in participants’ compliance to treatment. Among the health literacy variables, “Scale 3: Actively managing my health” (β = 0.38, *p* = 0.001) and “Scale 4: Social support for health” (β = 0.25, *p* = 0.019) had a significant effect on compliance to treatment. In this study, health literacy was identified as a key factor influencing compliance to treatment. Therefore, patients’ health literacy should be assessed prior to transplantation to identify potential high-risk patients for treatment nonadherence. In addition, after transplantation surgery, patient-tailored interventions should be developed and provided for self-management that reflects the patient’s health literacy level to ultimately enhance patient outcomes.

## 1. Introduction

Transplantation is a revolutionary method that has contributed to the prolongation of the lives of patients with organ failure. Advancements in medical technology in the past several decades—including the development of immunosuppressants—have improved the clinical outcomes for transplant recipients and increased their survival rates [1,2,3]. However, to ensure the long-term results of organ transplantation and prevent transplant failures, such as graft dysfunction, transplant recipients must be rigorously administered drugs, including immunosuppressants, throughout their life. Furthermore, they must ensure strict adherence to treatments that reduce various risk factors, ranging from infection and rejection to cardiovascular disease and liver failure [4,5,6].

Compliance is a term for closely following your doctor’s orders. Therefore, non-compliance refers to a state of not following the doctor’s instructions and orders, which can be harmful and fatal to individual patients and their families [7]. Noncompliance also leads to waste, as it reduces the potential benefits of therapy, and to the extra cost of treating avoidable consequent morbidity [7]. Compliance to treatment is a crucial health issue for transplantation patients in clinical practice [8]. However, the rate of noncompliance to treatment after transplantation tends to increase with time [6,9]. Compliance to treatment has also been reported to be closely associated with long-term graft survival and failure [6,10], resulting in continuous attempts to increase the rate of compliance to treatment by identifying its influencing factors. For example, if compliance to medication is poor, acute organ transplant rejection may occur, and it may be difficult to return to daily life with additional treatment and hospitalization [2,11]. In addition, poor adherence to dietary modification and sodium intake are important factors that are associated with poor outcomes after renal transplantation, including a higher risk of developing diabetes and cardiovascular mortality [12,13]. Despite its importance, as many as 15–40% of transplant recipients have demonstrated low compliance to treatment owing to immunosuppressant side effects or low literacy regarding new drugs [14,15].

Health literacy is defined as one’s ability to access, process, and understand health-related information and services to make good decisions regarding their health [16]. Inadequate compliance to treatment resulting from a poor understanding of medical information related to modified drug treatment after organ transplantation or the inability to search for suitable information regarding lifestyle correction or support systems increase the probability of re-hospitalization and transplant organ failure [17]. Therefore, health literacy has recently gained attention as a critical factor in compliance to treatment and health-related quality of life, which can enhance patients’ understanding of their health as well as their ability to follow the treatment regimens provided by health professionals after organ transplantation [16,18,19].

Transplant recipients experience various emotional changes after surgery. Most organ transplant recipients feel joy after having attained a new life through transplantation [20,21]; however, they also experience guilt and appreciation regarding the organ donor [21,22]. They may also worry about the side effects such as infections and concerns for cancer, damage to or rejection of the transplanted organ [21,22,23], and have mixed feelings about disclosing to others that they underwent organ transplantation surgery [21,22,23]. Thus, organ transplant recipients face various complicated emotional problems that may negatively affect their return to daily life after surgery along with compliance to treatment [7,24].

Various studies have attempted to identify the factors affecting compliance to treatment in transplantation patients, including investigations of patients’ disease-related knowledge and health literacy [25], depression [4], and self-efficacy [26]. Health literacy has been associated with patients’ educational level, intelligence, and age. Moreover, health literacy reportedly has an independent effect on compliance to treatment regardless of a high education level or long-term disease duration; this is because of an inadequate understanding of medical information, lack of appropriate access to information, and lack of support systems including health managers [17]. Therefore, it is necessary to determine the effects of health literacy on compliance to treatment in organ transplant recipients. Thus, this study aimed to determine the effects of health literacy and transplant on compliance to treatment in organ transplant recipients.

## 2. Materials and Methods

### 2.1. Study Design and Participants

A cross-sectional correlation design was used to determine the effects of health and transplant on compliance to treatment in organ transplant recipients. The participants were recruited from among organ transplantation outpatients who were regularly followed up with at the Department of Transplantation Surgery of a university hospital in Seoul, South Korea.

The inclusion criteria were as follows: (1) patients aged 19 years or older; (2) patients for whom one month or more had passed since the first transplantation surgery of the kidney or liver; (3) no psychiatric disorders, no health issues that may cause cognitive impairment (such as stroke or dementia), and no administration of CNS drugs; and (4) ability to communicate and provide responses to questionnaires. The exclusion criteria were as follows: (1) patients for whom the time passed since organ transplantation surgery was less than one year or who had undergone two or more organ transplantation surgeries; (2) patients who were below 19 years old; (3) patients with psychiatric disorders or a health issue that may cause cognitive impairment (such as stroke or dementia), individuals on CNS drugs; and (4) patients who were unable to communicate and provide responses to a questionnaire.

The sample size was estimated using the G-Power 3.1 program. At a significance level (α) = 0.05, power (1 − β) = 0.80, medium effect size = 0.15, and 11 significant predictors, a minimum of 121 participants were required; therefore, 130 patients were recruited, considering a dropout rate of approximately 10%. All the patients responded, so 130 patients were included in the final analysis.

### 2.2. Measures

#### 2.2.1. Health Literacy

The health literacy was measured using the self-reported Health Literacy questionnaire (HLQ) with proven validity and reliability. The HLQ was developed by Osborne et al. [16] specifically for surveys and intervention evaluations as well as for exploring the health literacy needs of both individuals and communities [16,27]. It consists of 44 questions in two parts and under nine subcategories. The Part 1 (from Scales 1 to 5) responses are rated on a 5-point Likert scale, ranging from 1 (strongly disagree) to 4 (strongly agree). Each subscale consists of four to six items. The Part 2 (from Scales 6 to 9) responses are rated on a 5-point Likert scale, ranging from 1 (cannot do or always difficult) to 5 (very easy). In this study, with the approval of the original HLQ author (License number #TL1505IA 1/1/2016), a Korean version was developed through translation and back-translation, according to the Translation Integrity Procedure (TIP) [16] developed by the original author. TIP is a systematic process of translation based on the translation management grid and item-intent format [16,27]. In the translation–back translation process, most of the forward translation content was accepted through a consensus derived from discussion. However, cultural characteristics were considered for certain items and the South Korean context was reflected through discussions with the original author (e.g., “I have all the information” was translated into “I feel there is information available to me”). In this study, the validity of the Korean version of the HLQ was verified for healthy Koreans [28]. The questionnaire does not include a total score. Instead, it has a mean domain-specific score calculated by adding each answer in a domain and dividing the score by the number of items in that specific domain. The scales have high and low descriptors to define the scope of the element of health literacy that the scale represents (Table 1) [29].

#### 2.2.2. Transplant Effects

The transplant effects were measured using the Transplant Effects Questionnaire (TxEO) developed by Ziegelmann et al. [30] based on literature reviews and focus group interviews to examine the response of transplant recipients to organ transplantation. The tool was translated by Kim [31] into a Korean version (K-TxEO), which was validated and verified in transplantation patients. The K-TxEO comprises 23 questions rated on a 5-point scale (from 1 = strongly disagree to 5 = strongly agree) with the following five subcategories: worry about the transplant organ (6 items), guilt regarding the organ donor (5 items), disclosure (3 items), treatment adherence (5 items), and responsibility toward family, health care team, and organ donor (4 items). A higher score indicates a higher level of transplant effects in the given subcategory, as perceived by the patient. In this study, treatment adherence was excluded and the remaining four subcategories of worry, guilt, disclosure, and responsibility were measured. The internal consistency reliability (Cronbach’s α) of the TxEQ at the time of its development was 0.72–0.86 [30], whereas the Cronbach’s α was 0.67–0.92 in the K-TxEO [31] and 0.67–0.90 in this study.

#### 2.2.3. Compliance to Treatment

The compliance to treatment was measured using an organ transplantation patients’ treatment compliance tool developed by Lee [32] to measure the compliance to treatment of kidney transplantation patients that was subsequently modified by Du [24]. The tool comprises 58 questions under nine subcategories: prevention of infection (9 items), drug administration (8 items), communication with the transplantation team (1 item), physical activities and exercise (1 item), diet (14 items), outpatient visits (4 items), general health management (8 items), other department visits (4 items), and emergency response (9 items). Each question is rated on a 4-point Likert scale ranging from 1 (never) to 4 (always). A higher score indicates higher compliance to treatment. The internal consistency reliability (Cronbach’s α) of compliance to treatment was 0.92 at the time of its development [32] and the Cronbach’s α was 0.94 in this study.

#### 2.2.4. Demographic and Disease-Related Characteristics

The demographic variables included sex, age, education, perceived economic status, type of donor, type of transplanted organ, time elapsed since transplantation, type of immunosuppressant prescribed, and side effects of immunosuppressants. All nine characteristics were assessed using a self-report questionnaire.

### 2.3. Data Collection

The data collection for this study was initiated after obtaining approval from the Institutional Review Boards of [AJIRB-SBR-SUR-18-122 and IRB-3-2019-0099], the authors’ affiliated university hospital, and the study site. This study was conducted in accordance with the principles of the Declaration of Helsinki.

The data collection was conducted from June to December 2019. The data were collected in cooperation with the Department of Transplantation Surgery at a university hospital in Seoul, South Korea. First, a trained research assistant obtained a list of outpatients from the Transplantation Surgery Department and selected participants who satisfied the inclusion and exclusion criteria. Next, each participant visiting the outpatient clinic on the scheduled date and received a leaflet describing the study purpose, methods, data privacy protection and management, consent to participate, and right to withdraw participation. This study only included participants who submitted signed consent forms. To ensure the autonomy and anonymity of the participants, each completed questionnaire was placed in a sealed envelope, which was dropped into a box. The collected questionnaires were encoded using codes identifiable only by the researcher. The time required to complete the questionnaire was approximately 20 min and the participants received a small gift as a token of appreciation.

### 2.4. Data Analysis

For the participant demographics and levels of health literacy, transplant effects, and compliance to treatment, the frequency, percentage, mean, and standard deviation were calculated. Regarding the variations in the compliance to treatment according to the participant demographics, an independent t-test and one-way ANOVA were performed and the Scheffe test was conducted as a post hoc test. To analyze the correlations between the health literacy, transplant effects, and compliance to treatment, the Pearson’s correlation coefficient was calculated. Finally, to identify the factors influencing the compliance to treatment, a multiple regression analysis was performed, including the variables analyzed as significant in the univariate analyses among the independent variables. Dummy variables were generated for the categorical variables as independent variables in the multiple regression analysis. All the collected data were analyzed using SPSS/WIN 25.0.

## 3. Results

### 3.1. Participants’ Demographics, Disease-Related Characteristics, and Differences in Compliance to Treatment

The demographic and disease-related characteristics of the participants are presented in Table 2. The percentage of males was 66.9%, with a mean age of 56.4 years. Most participants (51.5%) had an undergraduate degree or higher education level. Most (70.8%) perceived their economic status as “middle”. The percentage of living-donor transplantations was 73.8%, of which 66.9% were kidney transplantations. The mean duration after transplantation was 4.2 years (SD = 3.6), while 97.7% of participants were on two or more immunosuppressants and 20.8% had experienced an immunosuppressant-related side effect. The differences in compliance to treatment according to participant characteristics were statistically significant for sex (t = −2.17, *p* = 0.032) (Table 1).

### 3.2. Levels of Health Literacy, Transplant Effects, and Compliance to Treatment

In the Health Literacy Questionnaire (HLQ), the first five subscales (Scales 1–5) showed the highest score of 3.23 (SD = 0.48) for “Scale 4: Social support for health” and the lowest score of 2.67 (SD = 0.59) for “Scale 2: Having sufficient information to manage my health”. The remaining four subscales (Scales 6–9) showed the highest score of 3.42 (SD = 0.79) for “Scale 6: Ability to actively engage with health care providers” and the lowest score of 3.12 (SD = 0.87) for “Scale 8: Ability to find good health information”. Regarding the transplant effects among participants, the highest score was 4.13 (SD = 0.56) for responsibility, followed by 3.93 (SD = 0.82), 3.81 (SD = 1.04), and 3.06 (SD = 0.91) for worry, disclosure, and guilt, respectively. The compliance to treatment had a mean score of 4.36 (SD = 0.37) out of a total score of 5. The highest score of 4.70 (SD = 0.57) was found for “Communication with transplant team” and the lowest score of 4.18 (SD = 0.51) was found for “Diet” (Table 3).

### 3.3. Correlations among Health Literacy, Transplant Effects, and Compliance to Treatment

The correlations between health literacy, transplant effects, and compliance to treatment were analyzed (Table 4). All the health literacy variables, excluding “Scale 6: Active engagement with healthcare providers” and “Scale 7: Navigating the healthcare system”, showed a significant positive correlation with compliance to treatment (*p* < 0.05). Among the transplant effects, only responsibility had a significant positive correlation with compliance to treatment (*p* = 0.001).

### 3.4. Factors Influencing Compliance to Treatment

Prior to conducting the regression analysis, the basic assumptions of the regression analysis—such as residual independence, normality and homogeneity of variance, and multicollinearity—were tested, yielding satisfactory results. To identify the factors influencing the compliance to treatment, the demographic variables showing a significant variation and the health literacy and transplant effects variables were included as the independent variables in the multiple regression analysis. The model fit was F = 6.26 (*p* < 0.001) with 32.0% explanatory power. The results showed that, among the health literacy variables, “Actively managing my health” (β = 0.38, *p* = 0.001) and “Social support for health” (β = 0.25, *p* = 0.019) significantly influenced the compliance to treatment (Table 5).

## 4. Discussion

This cross-sectional study aimed to determine the level of compliance to treatment in patients who underwent organ transplantation and identify the factors that influence compliance to treatment. Thus, the study ultimately aimed to improve compliance to treatment in organ transplantation patients and provide self-reported findings from organ transplantation patients along with baseline data for developing intervention programs to enhance the quality of life of patients after organ transplantation.

Our findings showed that the mean score of the level of compliance to treatment in organ transplantation patients was 3.36 out of 4, which is high. This is consistent with the score of 3.26 reported by Du [24], who applied the same instrument to liver transplantation patients in South Korea. Nonetheless, international studies have reported that as many as 40% of organ transplantation patients do not strictly adhere to their treatment [14,15]. Compliance to treatment is a key predictor of prognosis in organ transplantation patients [6,10] and, hence, the education provided to patients emphasizes its importance and necessity. Among the subcategories of compliance to treatment, “diet” had a low score of 3.20. Similar to the general and other high-risk populations, transplant recipients must have a healthy diet and avoid excessive sodium intake [13]. However, the adherence to diet regimens was low because of low food literacy, fear of altering the diet, difficulties in adapting to a new diet, excessive appetite as a side effect of immunosuppressants, and lack of information regarding which foods to eat and from where to obtain diet facts [12]. Hence, to ensure a positive prognosis for the transplanted organ, the food literacy of transplantation patients should be assessed and improved by providing them with a tailored intervention.

Regarding the health literacy level in this study, across all nine subscales, the mean score was low for “Scale 2: Having sufficient information to manage my health” and “Scale 5: Appraisal of health information”. The low mean score on Scale 2 implies that the patient’s knowledge level is inadequate for resolving their health issues and managing their health; accordingly, the patient perceives a large gap between their health management knowledge and goals [27]. We presumed that Scale 2 exhibited the lowest mean score in this study because of the varying lengths of time elapsed since surgery; thus, the participants’ levels of experience regarding postoperative self-management and perceived knowledge varied. Although the knowledge level is not a component of health literacy, conceptual or disease-specific knowledge is a key resource for enhancing health literacy. Therefore, it is necessary to determine each patient’s knowledge and available resources according to the time elapsed since transplantation to provide continuous and tailored education on disease-specific knowledge.

Meanwhile, Scale 5 is concerned with the patient’s appraisal of health information [16] and a low score indicates that, despite their efforts, the patient cannot understand most of the available health information and feels confused in the face of contradictory data [27]. Scale 5 also demonstrated the lowest scores in previous studies [33,34] on health literacy in kidney transplant recipients. Poor scores on Scale 5 in transplantation patients may be owing to the low importance placed on the critical appraisal of health facts, even though patients are provided with detailed information across diverse areas—from drug administration to diet and exercise after transplantation. Nevertheless, it is possible to access health information through various media, including the Internet and social networking sites. Thus, patient education should focus on enhancing patients’ competence in selecting and appraising health information to assist in their postoperative adaptation and promote a successful transition.

For assessing transplant effects as a measure of transplantation-related experience [35], the highest and lowest scores were obtained for responsibility and guilt, respectively. Hence, organ transplant recipients strongly perceived responsibility toward their family, friends, and the transplantation team, but had the lowest perception of guilt for the donor. The sense of responsibility here represents feelings of wanting to take better care of oneself [30]. A high level of responsibility may correlate with compliance to treatment. In South Korea, there is a strong collectivist culture and a sense of duty to family; therefore, living donor transplantation is more common than deceased donor transplantation [31]. This may account for the high level of responsibility toward others experienced by Korean patients after transplantation. To identify the emotional responses that promote compliance to treatment in organ transplant recipients, a qualitative study should be conducted in the future to investigate feelings of guilt toward donors and responsibility toward others, as perceived by transplant recipients. Additionally, the relationship between these two factors should be compared.

In this study, the multiple regression analysis revealed that, among the subscales of health literacy, “Scale 3: Actively managing my health” and “Scale 4: Social support for health” influenced the compliance to treatment in organ transplantation patients. Notably, “Scale 3: Actively managing my health” was identified as the strongest influencing factor of compliance to treatment in organ transplantation patients. Scale 3 is concerned with self-assessment of health management and a high score indicates that the patient is actively managing their health with a sense of responsibility for health maintenance and management [16]. Thus, a higher score on Scale 3 in this study resulted in a higher level of compliance to treatment. To improve both health management literacy and compliance to treatment, it is necessary to increase patients’ feelings of responsibility for their health maintenance and management by including them in their decision-making regarding the treatment process and health management.

“Scale 4: Social support for health” was identified as another influencing factor of compliance to treatment in organ transplantation patients. These patients must continuously adhere to healthy behaviors, including a healthy diet and drug therapy, even after surgery. Thus, they should be able to find and apply relevant health information or health management systems to assist with their health management. If the patient is unable to achieve this, then their family or friends may adopt the role of decision-maker and apply for health care services on behalf of the patient [33]. Therefore, the higher score obtained for Scale 4 in this study is considered to have led to a higher-level compliance to treatment. The health care team should determine in advance the patient’s levels of social support and perceived social support for health to screen for a high-risk group of treatment non-adherence and, when necessary, request collaboration with various support teams.

While evidence to support the direct impact of health literacy on clinical results remains insufficient, health literacy has been established as a critical factor of self-management. Health literacy has been previously applied in the simple measurements of patients’ cognitive functions or intelligence as a factor related to the potential ability of organ transplantation patients and patients with chronic disease to perform self-management [36]. However, the HLQ allows for the evaluation of patients’ abilities to search for information regarding self-management, to explore knowledge, and to recognize their health issues and consult a health manager. Therefore, the HLQ is viewed as an effective tool for assessing the demand for personalized interventions in clinical practice. Hence, the HLQ was used in this study to determine the level of health literacy in organ transplantation patients and the results are significant as we identified two important influencing factors of compliance to treatment: “actively managing my health” and “social support for health”. To enhance the compliance to treatment in organ transplantation patients, their literacy regarding health and health management should be determined and the healthcare team should endeavor to improve patient literacy beginning from the time the surgery is planned.

However, the results of this study should be interpreted carefully because of the following limitations: First, the generalizability of this study’s findings is limited because it was conducted solely among registered organ transplantation patients at a single transplantation center. Second, the compliance to treatment was assessed using a self-reported questionnaire rather than by directly monitoring each patient. Thus, the observed and actual levels of compliance to treatment may vary only slightly. Finally, the Korean version of the HLQ was used. The K-HLQ was developed through translation–back translation, in accordance with the systematic TIP, and was finalized after revision and complementation based on discussions with the original developer. However, there may be limitations due to the lack of psychosocial assessments of transplantation patients in South Korea and of the simultaneous assessments of participants’ cognitive functions and intelligence. Therefore, follow-up studies should conduct simultaneous measurements of the objective indicators of compliance to treatment and health literacy. Furthermore, the effect of health literacy on compliance to treatment should be accurately analyzed through a longitudinal study to define the trajectory of variation in compliance to treatment among target patients. 

## 5. Conclusions

It is therefore necessary to determine the effects of health literacy compliance to treatment in organ transplant recipients. Thus, this study aimed to determine the effects of health literacy and transplant effects on the compliance to treatment in organ transplant recipients.

## Figures and Tables

**Table 1 jcm-12-00977-t001:** Health Literacy Questionnaire scales and descriptors.

Scale Number and Name	Interpretation
1. Feeling understood and supported by healthcare providers	High: Has an established relationship with at least one healthcare provider who knows them well and who they trust to provide useful advice and information and to assist them to understand information and make decisions about their health. Low: People who are low on this domain are unable to engage with doctors and other healthcare providers.
2. Having sufficient information to manage my health	High: Feels confident that they have all the information that they need to live with and manage their condition and to make decisions. Low: Feels that there are many gaps in their knowledge and that they do not have the information they need to live with and manage their health concerns.
3. Actively managing my health	High: Recognize the importance of and are able to take responsibility for their own health. They proactively engage in their own care and make their own decisions about their health. Low: People with low levels do not see their health as their responsibility, they are not engaged in their healthcare and regard healthcare as something that is performed on them.
4. Social support for health	High: A person’s social system provides them with all the support they want or need. Low: Completely alone and unsupported.
5. Appraisal of health information	High: Able to identify good information and reliable sources of information. Low: No matter how hard they try, they cannot understand most health information and become confused when there is conflicting information.
6. Ability to actively engage with healthcare providers	High: Is proactive about their health and feels in control in relationships with healthcare providers. Is able to seek advice from additional health care providers when necessary. Low: Is passive in their approach to health care, inactive. They accept information without question. Unable to ask questions to receive information or to clarify what they do not understand. Feel unable to share concerns.
7. Navigating the healthcare system	High: Able to find out about services and supports so they have all their needs met. Low: Unable to advocate on their own behalf and unable to find someone who can help them use the healthcare system to address their health needs.
8. Ability to find good health information	High: Is an ‘information explorer’. Actively uses a diverse range of sources to find information and is up to date. Low: Cannot access health information when required. Is dependent on others to offer information.
9. Understand health information well enough to know what to do	High: Is able to understand all written information (including numerical information) in relation to their health and able to write appropriately on forms where required. Low: Has problems understanding any written health information or instructions about treatments or medications.

**Table 2 jcm-12-00977-t002:** Differences of compliance to treatment according to general characteristics (*n* = 130).

Variables	Categories	*n*(%) or Mean ± SD	Compliance to Treatment	T or F (*p*)
			Mean ± SD	
Gender	Male	87 (66.9)	3.32 ± 0.41	−2.27 (0.025)
	Female	43 (33.1)	3.45 ± 0.28	
Age	≤54	53 (40.8)	3.36 ± 0.38	0.03 (0.970)
	55–64	43 (33.1)	3.37 ± 0.40	
	≥65	34 (26.1)	3.35 ± 0.34	
		56.4 ± 11.6		
Education	≤Middle school	21 (16.2)	3.41 ± 0.37	0.22 (0.803)
	High school	42 (32.3)	3.37 ± 0.41	
	≥College	67 (51.5)	3.35 ± 0.35	
Perceived economic status	High	18 (13.8)	3.33 ± 0.94	0.10 (0.909)
	Middle	92 (70.8)	3.37 ± 0.04	
	Low	20 (15.4)	3.38 ± 0.08	
Types of donors	Living	96 (73.8)	3.38 ± 0.38	0.96 (0.337)
	Deceased	34 (26.2)	3.31 ± 0.35	
Type of transplanted organ	Kidney	87 (66.9)	3.37 ± 0.37	0.22 (0.828)
	Liver	43 (33.1)	3.35 ± 0.39	
Duration after transplantation (y)	<1	33 (25.4)	3.36 ± 0.37	2.27 (0.107)
	1–5	47 (36.2)	3.27 ± 0.37	
	≥5	50 (38.6)	3.43 ± 0.37	
		4.2 ± 3.2		
Types of prescribed immunosuppressants	1	3 (2.3)	3.28 ± 0.70	0.15 (0.862)
	2	81 (62.3)	3.37 ± 0.38	
	3	46 (35.4)	3.34 ± 0.35	
Side effect of immunosuppressants	Yes	27 (20.8)	3.35 ± 0.37	−0.19 (0.848)
	No	103 (79.2)	3.37 ± 0.37	

**Table 3 jcm-12-00977-t003:** Health literacy, transplant effects, and compliance to treatment (*n* = 130).

Variables	Min	Max	Mean ± SD
Health literacy			
Scale 1: Feeling understood and supported by healthcare providers	1.00	4.00	3.07 ± 0.54
Scale 2: Having sufficient information to manage my health	1.00	4.00	2.67 ± 0.59
Scale 3: Actively managing health	1.00	4.00	2.89 ± 0.58
Scale 4: Social support for health	1.00	4.00	3.23 ± 0.48
Scale 5: Appraisal of health information	1.20	4.00	2.81 ± 0.52
Scale 6: Ability to actively engage with health care providers	1.00	5.00	3.42 ± 0.79
Scale 7: Navigating the healthcare system	1.00	5.00	3.22 ± 0.86
Scale 8: Ability to find good health information	1.00	5.00	3.12 ± 0.87
Scale 9: Understand health information well enough to know what to do	1.00	5.00	3.31 ± 0.80
Transplant effects			
Disclosure	1.00	5.00	3.81 ± 1.04
Guilt	1.00	5.00	3.06 ± 0.91
Worry	1.67	5.00	3.92 ± 0.82
Responsibility	2.67	5.00	4.13 ± 0.56
Compliance to treatment	2.36	4.00	3.36 ± 0.37
Infection prevention	2.22	4.00	3.51 ± 0.41
Take medication	2.50	4.00	3.59 ± 0.38
Communication with transplant team	1.00	4.00	3.70 ± 0.57
Activity and exercise	1.00	4.00	3.48 ± 0.83
Diet	1.64	4.00	3.20 ± 0.50
Outpatient visit	2.25	4.00	3.40 ± 0.44
General health care	1.33	4.00	3.19 ± 0.76
Other outpatient visits	1.89	4.00	3.29 ± 0.47
Emergency response	1.00	4.00	3.33 ± 0.73

**Table 4 jcm-12-00977-t004:** Correlation between health literacy, transplant effects, and treatment compliance (*n* = 130).

Variables	Compliance to Treatment
	r (*p*)
Health literacy	
Scale 1: Feeling understood and supported by healthcare providers	0.221 (0.012)
Scale 2: Having sufficient information to manage my health	0.263 (0.003)
Scale 3: Actively managing health	0.262 (0.003)
Scale 4: Social support for health	0.352 (<0.001)
Scale 5: Appraisal of health information	0.465 (<0.001)
Scale 6: Ability to actively engage with health care providers	0.166 (0.059)
Scale 7: Navigating the healthcare system	0.168 (0.056)
Scale 8: Ability to find good health information	0.206 (0.019)
Scale 9: Understand health information well enough to know what to do	0.243 (0.005)
Transplant effects	
Disclosure	0.069 (0.437)
Guilt	−0.032 (0.716)
Worry	0.064 (0.468)
Responsibility	0.294 (0.001)

**Table 5 jcm-12-00977-t005:** Factors associated with compliance to treatment (*n* = 130).

Variables	B	SE	β	t	*p*
Gender (Female)	0.12	0.06	0.15	1.89	0.061
Health literacy					
Scale 1: Feeling understood and supported by healthcare providers	−0.13	0.08	−0.19	−1.73	0.087
Scale 2: Having sufficient information to manage my health	−0.08	0.08	−0.13	−1.02	0.311
Scale 3: Actively managing health	0.25	0.07	0.38	3.53	0.001
Scale 4: Social support for health	0.20	0.08	0.25	2.39	0.019
Scale 5: Appraisal of health information	0.08	0.09	0.12	0.94	0.347
Scale 8: Ability to find good health information	−0.03	0.06	−0.07	−0.51	0.610
Scale 9: Understand health information well enough to know what to do	0.08	0.06	0.17	1.28	0.203
Transplant Effects					
Responsibility	0.08	0.06	0.12	1.42	0.157
R^2^ = 0.32, Adjusted R^2^ = 0.27, F = 6.26, *p <* 0.001					

## Data Availability

Not applicable.

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
