# Peer review of "A Cross-Sectional Analysis of Health Literacy and Compliance to Treatment in Organ Transplant Recipients"

_jcm, 2023, doi:10.3390/jcm12030977_

Round 1
Reviewer 1 Report
Overall, this was a well written manuscript.
The Abstract although was valid in the points of outlining that “health literacy should be assessed prior to transplantation to identify potential high-risk patients for treatment nonadherence” and “patient-tailored interventions should be developed and provided for self-management that reflects the patient’s health literacy level to ultimately enhance patient outcomes” these points were not taken to the next level of consideration such as utilizing the research findings to describe next steps for implementation to support these statements.
In regards to “Transplant recipients experience various emotional changes after surgery” the gaps identified in this paragraph included not stating specific side effects such as infections and concern for cancer, which are not trivial for patients. An additional gap was the notion of treatment adherence based on the frequency of immunosuppressant medications and adjunct therapy interfering or impeding ADLs and Return to Work. Although ADLs was mentioned in this section, it was in a different context.
It is confusing to state, that “Considering the dropout rate, the questionnaire 91 was distributed to 130 individuals, and all 130 questionnaires (100.0%)” given all of the 130 questionnaires were completed. Consider revising this sentence.
Very interesting finding regarding, that “among the transplant 212 effects, only responsibility had a significant positive correlation with treatment adherence.” The likely justification for this was described very well in the Discussion section as "a strong collectivist culture and a sense of duty to family.
Dietary modifications and sodium intake are very important in this patient population; however, seemed like an afterthought since only briefly mentioned prior to the Discussion section; however, was really highlighted as a themed subcategory of treatment adherence. Consider highlighting this earlier in the manuscript.
In the section "Factor influencing treatment adherence", the only two parameters reaching significance were "actively managing my health" and "social support for heath". Both were deemed to be "sufficiently influencing treatment adherence". I wonder if this really fits into the definition of "literacy". My definition of literacy is "competence of knowledge in a specific area". Is not being good at managing own heath or not have sufficient social support due to literacy ? Can an educated 25 years old patient managing his/hers own health be not compliant because d maturity rather than literacy ? Should the title be adjusted to include social parameters. Please expand on this.
In the Discussion section it is worth distinguishing again that this is based on abdominal organ transplants, you do however, note the limitation of generalizability based on a single center, but you should also note a limitation being due to a subset of abdominal organ transplant patients versus all solid organ transplant patients. That should also be further noted in the conclusion.
Author Response
Reviewer 1
The Abstract although was valid in the points of outlining that “health literacy should be assessed prior to transplantation to identify potential high-risk patients for treatment nonadherence” and “patient-tailored interventions should be developed and provided for self-management that reflects the patient’s health literacy level to ultimately enhance patient outcomes” these points were not taken to the next level of consideration such as utilizing the research findings to describe next steps for implementation to support these statements.
-> Thank you for your comments. Based on your comment, I've deleted the sentences in lines 22-24 for clarity and better understanding for readers.
In regards to “Transplant recipients experience various emotional changes after surgery” the gaps identified in this paragraph included not stating specific side effects such as infections and concern for cancer, which are not trivial for patients. An additional gap was the notion of treatment adherence based on the frequency of immunosuppressant medications and adjunct therapy interfering or impeding ADLs and Return to Work. Although ADLs was mentioned in this section, it was in a different context.
-> Based on your opinion, we have added and supplemented the contents of the main text in line 44-49, and 66-67.
It is confusing to state, that “Considering the dropout rate, the questionnaire was distributed to 130 individuals, and all 130 questionnaires (100.0%)” given all of the 130 questionnaires were completed. Consider revising this sentence.
-> We revised the sentence to improve the clarity of the text. Please check line 102-1043.
Very interesting finding regarding, that “among the transplant effects, only responsibility had a significant positive correlation with treatment adherence.” The likely justification for this was described very well in the Discussion section as "a strong collectivist culture and a sense of duty to family.
Dietary modifications and sodium intake are very important in this patient population; however, seemed like an afterthought since only briefly mentioned prior to the Discussion section; however, was really highlighted as a themed subcategory of treatment adherence. Consider highlighting this earlier in the manuscript.
-> Thank you for your considerable comment. According to your comments, the manuscript was modified. Please check the line 46 -48.
In the section "Factor influencing treatment adherence", the only two parameters reaching significance were "actively managing my health" and "social support for heath". Both were deemed to be "sufficiently influencing treatment adherence". I wonder if this really fits into the definition of "literacy". My definition of literacy is "competence of knowledge in a specific area". Is not being good at managing own heath or not have sufficient social support due to literacy ? Can an educated 25 years old patient managing his/hers own health be not compliant because d maturity rather than literacy ? Should the title be adjusted to include social parameters. Please expand on this.
-> Thank you for your considerable comment. To measure health knowledge in this study, we used a tool developed to measure “one’s ability to access, process, and understand health-related information and services to make good decisions regarding their health.” However, in order to help readers' understanding, the characteristics of the sub-domains of the Korean version of the HLQ tool were added and described. Please check the line 129-131.
In the Discussion section it is worth distinguishing again that this is based on abdominal organ transplants, you do however, note the limitation of generalizability based on a single center, but you should also note a limitation being due to a subset of abdominal organ transplant patients versus all solid organ transplant patients. That should also be further noted in the conclusion.
-> Thank you for your comments. Added limitation that it only deals with abdominal organ transplantation in line 460.
Reviewer 2 Report
Thank you very much for the opportunity to review the manuscript titled: A Cross-sectional Analysis of Health Literacy and Treatment Adherence in Organ Transplant Recipients
The more accurate will be adherence to treatment than treatment adherence.
Please define the term "nonadherence" in the introduction section.
Please give the name of the questionnaire measuring treatment adherence.
Author Response
Reviewer 2
Thank you very much for the opportunity to review the manuscript titled: A Cross-sectional Analysis of Health Literacy and Treatment Adherence in Organ Transplant Recipients
The more accurate will be adherence to treatment than treatment adherence.
-> Thanks for your comments. This study attempted to identify the factors affecting the compliance and treatment regimen of organ transplant patients. Accordingly, 'treatment adherence' was modified to 'treatment compliance', and the definition of 'treatment noncompliance' was added to the text (Please check Line 35-39).
Please give the name of the questionnaire measuring treatment adherence.
-> We added the name of the questionnaire measuring treatment adherence in line 102-104.